# Preconception Care among Pregnant Women in an Urban and a Rural Health Facility in Kenya: A Quantitative Study

**DOI:** 10.3390/ijerph17207430

**Published:** 2020-10-13

**Authors:** Joan Okemo, Marleen Temmerman, Mukaindo Mwaniki, Dorothy Kamya

**Affiliations:** 1Department of Obstetrics ad Gynecology, Aga Khan University Hospital, Nairobi 30270-00100, Kenya; marleen.temmerman@aku.edu (M.T.); mukaindo.mwaniki@aku.edu (M.M.); 2Department of Postgraduate Medical Education, Aga Khan University Hospital, Nairobi 30270-00100, Kenya; dorothy.kamya@aku.edu

**Keywords:** preconception care, Kenya, urban, rural, level, determinants, utilization

## Abstract

Preconception care (PCC) aims to improve maternal and fetal health outcomes, however, its utilization remains low in developing countries. This pilot study assesses the level and determinants of PCC in an urban and a rural health facility in Kenya. Unselected pregnant women were recruited consecutively at the Mother and Child Health (MCH) clinics in Aga Khan University Hospital, Nairobi (AKUH, N-urban) and Maragua Level Four Hospital (MLFH-rural). The utilization of PCC was defined as contact with any health care provider before current pregnancy and addressing pregnancy planning and preparation. A cross-sectional approach was employed and data were analyzed using SPSS version 22. 194 participants were recruited (97 in each setting) of whom, 25.8% received PCC. Age, marital status, education, parity and occupation were significant determinants of PCC uptake. There was also a significant difference in PCC uptake between the rural (16.5%) and urban (35.1%) participants (*p* < 0.01), OR of 0.3 (0.19–0.72, 95% CI). The low level of PCC in Kenya revealed in this study is consistent with the low levels globally. However, this study was not powered to allow firm conclusions and analyze the true effects of PCC determinants. Therefore, further research in the field is recommended in order to inform strategies for increasing PCC utilization and awareness in Kenya.

## 1. Introduction

The World Health Organization (WHO) defines preconception care (PCC) as the provision of biomedical, behavioral and social health interventions to women and couples before conception occurs, with the aim of improving their health status, and mitigating behaviors, individual and environmental factors that could contribute to poor maternal and child health outcomes [1,2]. This is done through risk identification, health education and promotion and initiation of evidence-based interventions in the period prior to conception. The use of PCC use in high- and in low-income countries aims to improve maternal pregnancy and neonatal outcomes both in the short and long term [3].

The Ministry of Health in Kenya recommends PCC as one of the pillars aimed at attaining the fourth and fifth Millennium Development Goals (MDGs) that aim to reduce child mortality and improve maternal health respectively [4]. These MDGs were followed by the Sustainable Development Goals (SDGs), where SDG 3 aims to ensure healthy lives and promote well-being for all at all ages by 2030 [5]. The Kenya National Reproductive Health System (KNRHS) of 2009–2015 [4] recognizes PCC and family planning as one of the pillars of maternal and newborn health. Others include focused antenatal care (ANC), essential obstetric care, essential newborn care, targeted postpartum care and post-abortion care. These were deemed essential in accomplishing MDGs four and five [6]. While the other pillars had laid out strategies for implementation, there is a paucity of data with regard to how PCC was to be implemented nationwide.

Several studies conducted to look at the level of PCC in other parts of the world found the levels to be generally low. A study conducted by Frey and Files in Mayo clinic [7] found that only 39% of the women received PCC from their primary care physicians compared to 98.6% who believed in its importance. Ezegwui et al. [8] in Southeastern Nigeria found that 43.1% of their study participants were aware of PCC and of these, 64.4% had correct knowledge. Only 32.6% of the aware group (14% overall) involved a health practitioner during the planning of their pregnancies despite a good percentage having a planned pregnancy (68.4%). Stephenson et al. [9] found that 27% of the women in their study sought advice from health providers about getting pregnant despite a 73% rate of planned pregnancies.

Some studies in Africa have demonstrated a difference in the level of PCC among women in urban and rural settings. Ezegwui et al. [8] and Ekem et al. [10] in Southeastern Nigeria conducted their studies in urban settings and found the level of PCC awareness to be 43.1% and 44.2%, with a 14% and 10.3% rates of utilization respectively. A different study by Idris et al. [11] in Northwestern Nigeria in a semi-urban setting found only 4% of the 150 recruited women were aware of PCC and the level of utilization was 2.7%.

While there were no studies identified in the literature that looked at the level and determinants of PCC in Kenya, the statistical differences in the level of other forms of maternal health services in urban and rural Kenya may imply differences in PCC as well. In Kenya, 4% of pregnant women do not seek ANC at all (5.3% in rural versus 1.8% in urban) [12]. According to KDHS 2014 [12], contraceptive prevalence rate (CPR) was 62% in urban and 56% in rural areas. Efforts to increase CPR in Kenya have been met by challenges such as poverty, religious, cultural beliefs and practices and weak health systems [4]. Other statistical differences were: level of unmet contraceptive need of 13% (urban) and 20% (rural); women who received ANC from a skilled health provider were 97.8% (urban) and 94.0% (rural) and women who sought postnatal care were 65% (urban) and 42.7% (rural).

Our study aimed to compare the level of PCC utilization and its determinants among pregnant women in urban (Aga Khan University Hospital, Nairobi-AKUH, N) and rural (Maragua Level Four Hospital-MLFH) settings in Kenya.

## 2. Materials and Methods

The study’s definition of PCC utilization was contact with any health provider before current pregnancy and having discussed pregnancy planning and preparation. To elicit the level of PCC utilization, we used question 3 in section four of the questionnaire which stated, “I talked about pregnancy plans and preparation with a health care provider before I got pregnant”.

An analytic cross-sectional study was used to compare the level of PCC in AKUH, N and MLFH and some of the factors affecting PCC through a self-administered questionnaire. The study was conducted at the AKUH, N and MLFH Mother–Child Health Clinics (MCH).

AKUH, N is an urban private, tertiary, teaching and referral health facility located in Nairobi County—an all urban county [13]. It is an academic health care centre providing tertiary level healthcare. Aga Khan Hospital was founded in 1955 and became part of the Aga Khan University in 2005 which evolved it into a premier teaching and tertiary care referral hospital. AKUH, N has a bed capacity of 254 and offers state of the art services including maternity and antenatal services in the MCH clinic. From the AKUH, N maternity records, about 800 women are seen in the MCH clinic monthly and on average, 350 deliveries are conducted monthly in the maternity ward with approximately 45% of these being caesarian deliveries.

MLFH is a public hospital located about one kilometer from Maragua town in the southern part of Murang’a County [14]. Level four hospitals are the first referral hospitals in Kenya, with both outpatient and inpatient health services and referral services. They have the following clinical services which are run by either an on-site specialist in the field or a specialist who covers several of the level four hospitals within the region: obstetrics and gynaecology; child health; medicine; surgery and anaesthesia. Murang’a County has a dense rural settlement with 89% of the population living in rural areas and only 11% living in urban centres. Maragua’s constituency is largely rural. Most of the road linkages within the county are all-weather roads with some of the economic activities including farming, quarrying, forestry and tourism [14]. MLFH was founded in 1972 as a rural health training and demonstration centre for Kenya Medical Training College by then Minister James Njiru. It became a level four hospital for Murang’a South in 1997. It offers several services including antenatal services through the MCH clinic that serves over 400 pregnant women monthly. It has a bed capacity of 82 and conducts about 350 deliveries monthly with a 30% caesarian section rate.

The study population included all pregnant women attending antenatal care in one of the two hospitals. The participants were eligible if they were aged ≥18 years and were able to speak English or Kiswahili. The KDHS 2014 report [12] indicates that over 97% of pregnant women in the two counties have at least one contact with a skilled health provider. Obtaining study subjects from the population of pregnant women attending ANC in both hospitals provided a good representation of the pregnant women population and by extension, reproductive-aged women.

Due to a lack of locally published prevalence studies on PCC, we based our sample size calculation on Nigerian data using a difference in PCC prevalence of 14% in urban and 2.7% in semi-urban by Ezegwui et al. and Idris et al. respectively [8,11]. We calculated the sample size needed to do the study with a power of 80% and a confidence interval of 95% as 97 women in each site. The formula for difference in two proportions shown below was used for sample size calculation,
$$n=\;2\;\times \;(Z\alpha \;+\;Z\beta {)}^{2}\;\times \;\frac{P1(1\;-\;P1)\;+\;P2(1\;-\;P2)}{{\left(P1\;-\;P2\right)}^{2}}$$
where,

*n* = sample size in each group (equal sample size in each group).

P1 = estimated proportion of study outcome in the AKUH, N group (urban). In this case, −14% as derived from the Nigerian study by Ezegwui et al. [8].

P2 = estimated proportion of the study outcome in the Maragua group (rural). In this case, 2.7% as derived from the Nigerian study by Idris et al. [11].

Zα = critical value at the level of statistical significance (1.96)

Zβ = critical value at the level of the desired power (0.84 for 80% power).

The data collection tools were in both English and Kiswahili languages. The tools were developed based on the study objectives and literature review. They were thereafter validated on 8 antenatal women in AKUH, N and MLFH MCH clinics to test the idioms of the languages used, response time, check whether the questions were appropriately framed, inoffensive, clear, easy to understand, able to elicit discussions and address the intended questions for the study. They were found to be suitable for the study and did not require any adjustments.

Data collection was conducted during the waiting period before consultation with the health providers at the respective MCH clinics. A 5–10 min self-administered questionnaire was used to collect data, including the socio-demographic section (age, marital status, education, occupation, residential area); obstetric history (parity, gestational age in weeks by last menstrual period (LMP) or first-trimester scan and prior obstetrics outcomes); pre-existing medical conditions and a section with questions to determine the level of PCC prior to current pregnancy and some more determinants.

Participants were selected through consecutive sampling. All eligible participants who met the inclusion criteria were approached and explained the purpose of the study by the principal investigator and two research assistants. A hundred women were approached in AKUH, N of whom 97 consented to participate in the study and 3 declined. A hundred and one women were approached in MLFH, of whom 97 consented to participate in the study and 4 declined. The willing participants chose their preferred language (English or Kiswahili) and were then required to sign an informed consent form. Thereafter, a 5–10 min self-administered questionnaire was given for the collection of data on socio-demographics, obstetrics and clinical history, level and some determinants of PCC utilization. The questionnaires were confirmed to be completely filled by the research assistants/principal investigator before a participant left. The same processes of data collection were carried out in AKUH, N and MLFH.

Data were analyzed using SPSS version 22. Descriptive analysis was done to describe the characteristics of the study participants. Binary variables were constructed to conduct further univariate analysis. The Chi-square test and Student’s *t*-test were used to test for the association for the categorical and continuous variables respectively. A *p*-value of <0.05 was considered significant. Unadjusted odds ratios were calculated and presented with 95% confidence intervals. Variables whose *p*-values were found to be significant from the univariate analysis were subjected to further multivariate analysis through the calculation of adjusted odds ratios in order to remove the effects of potential confounders. However, because of the small study sample size, firm conclusions could not be drawn from this analysis.

All subjects gave their informed consent for inclusion before they participated in the study. The study was conducted in accordance with the Declaration of Helsinki, and the protocol was approved by the Aga Khan University Hospital Nairobi Research and Ethics Committee. The corresponding ethical approval code is Ref: 2016/REF-61(v2) dated 16 February 2017.

## 3. Results

### 3.1. Study Participants

Table 1 summarizes the characteristics of the study participants. The two groups were comparable for parity, prior pregnancy outcomes, caesarian section deliveries, medical conditions during prior pregnancy and preexisting medical conditions. However, there were significant differences in age, marital status, educational level and occupation (*p* < 0.05).

Table 2 summarizes the health-seeking behaviour of study participants by site and level of significance. There were differences in visits to health professionals, challenges of access to health care, pregnancy timing, beliefs in ability to prepare for pregnancy and opinions on the availability of PCC information (*p* < 0.05).

Women who received PCC in both sites were 25.8%. There was a significant difference (*p* < 0.01) in the level of PCC in AKUHN, 35.1% compared to MLFH, 16.5%.

### 3.2. Uptake and Determinants of PCC

A total of 50 out of the 194 women (25.8%) received PCC. Of these, 34 women were from AKUH, N and 16 were from MLFH. In the AKUH, N group, 35.1% received PCC as opposed to 16.5% from the MLFH group. There was a significant difference in the level of PCC between the two study sites with a *p*-value of <0.01. The odds ratio was found to be 0.3 (0.2–0.7) which means that the odds of receiving PCC in the MLFH group were 70% less than in the AKUH, N group. These results have have also been captured in a video summary (Appendix A). Figure 1 below demonstrates the level of PCC in the two study sites.

Table 3 shows the results from the univariate analysis of the association between PCC and other study variables. Age, site, marital status, education, occupation, parity, prior pregnancy outcomes, seeking PCC elsewhere other than from health providers and pregnancy timing were significantly associated with PCC.

Univariate analysis revealed a strong association between PCC and age, site, marital status, education level, occupation, parity, prior poor pregnancy outcomes, seeking PCC elsewhere and pregnancy timing. These variables were entered into SPSS version 22 and subjected to multivariate logistic regression with the uptake of PCC being the dependent variable and the others (age, site, marital status, education level, occupation, parity, prior poor pregnancy outcomes, seeking PCC elsewhere and pregnancy timing) as independent variables, in order to control for the influence of each of the independent variables. The independent variables were entered together in SPSS and not one by one. This analysis generated adjusted odds ratios which were presented with a 95% confidence interval as outlined in Table 4 below. However, because of the low sample size the confidence intervals were wide and no significant difference below the 0.05 level were found, but trends towards statistical significance were observed for education and prior poor pregnancy outcomes.

## 4. Discussion

In this study, only 25.8% of the participants received PCC while 96% of pregnant women in Kenya are attending ANC [5]. This points to different factors affecting the utilization of different maternal health services in Kenya. From this study, many women felt that there was not enough information about PCC which could partly explain its underutilization locally. However, our study revealed that lack of information on PCC was not a significant determinant of PCC utilization pointing to a role that other factors play. Factors such as age, marital status, site of health facilities, parity and prior poor pregnancy outcomes were shown to have potential effects on PCC utilization. Studies that have looked at the determinants of other maternal health services in Kenya [15,16,17] and determinants of PCC utilization in other parts of the world [8,9,18] demonstrated the role played by the aforementioned factors. The older, married, parous women as well as those who reside in urban areas and those with a history of prior poor pregnancy outcome, are more likely to use PCC as well as other maternal health services which agrees with what was demonstrated in this study. Our findings align with the low uptake of PCC in Africa and globally in comparison to other maternal health services [7,8,9,10,11,18], underlining the need to create awareness and demand for PCC both locally and globally.

More women in the urban health facility received PCC compared to the rural health facility which could be due to differences in age, marital status, education, occupation and parity between the sites as shown by earlier data from Kenya. Several studies in Kenya showed that older maternal age, higher education level, being employed and being married—especially marriage to an educated and employed partner—were significantly associated with the increased use of maternal health services, while parity had mixed effects [15,16,17]. The mixed effects of parity could arise from prior use of maternal services that could lead to feeling well versed hence no further need for them or having insight into the importance of the services hence encouraging more use.

The mean age at the urban facility was higher than at the rural facility. This is in keeping with the trend of higher maternal age in urban areas compared with the rural areas as women pursue education and career advancement and postpone marriage [12]. In support of this, there were more women with tertiary education and formal employment in the urban health facility. These work hand in hand to increase both awareness and utilization through increasing financial accessibility. Additionally, more women in the urban health facility believed there was enough PCC information and fewer had financial challenges. The proportion of married women in the sample differed at the urban and rural sites—with more married women in the urban sample. It is, therefore, possible that the older, more educated hence more informed, employed and married women in urban settings were advantaged and hence likely to receive PCC compared to their rural counterparts. This can also be supported by the higher rates of routine PCC in the urban health facility.

Another reason for the difference in PCC could arise from the higher rates of the urban women who had their pregnancies planned, which may imply more time, thought and deliberation put into the process, and hence the likelihood of seeking help to optimize pregnancy outcomes. Family planning is part and parcel of PCC as it may help to optimize maternal health prior to pregnancy, for example in the setting of chronic and infectious diseases. This was supported by findings from other studies in the literature of pregnancy planning as a promoter of PCC and vice versa [9,18]. This is further supported by the KDHS 2014 report that showed a higher CPR and lower unmet contraceptive need in urban compared to rural areas [12]. The PCC difference could also be explained by the higher number of women in the urban setting who believed in the possibility of women preparing for pregnancy, which may translate to a higher self-efficacy, and consequently, more receptivity to PCC services.

In line with what would be expected, there were more women in the urban setting with poor prior pregnancy outcomes despite having a higher level of PCC. A possible explanation is that the higher number of women with poor prior pregnancy outcomes in the urban setting served as the driving force for more PCC utilization in pursuit of better outcomes in the future. Overall, PCC provides an opportunity to optimize a woman’s health in order to provide a safe fetal environment and consequently offer multigenerational benefits. It also results in improved health of the children through prevention of infection transmission such as HIV/AIDS (human immunodeficiency virus and acquired immune deficiency syndrome) and lowering the risk of some forms of childhood cancers, obesity, type 2 diabetes mellitus and cardiovascular diseases in later life [1,19]. Alternatively, AKUH, N being a tertiary and referral hospital, could be receiving more women referred there due to bad prior outcomes compared to MLFH, a public level four hospital. Another explanation could be that some women in MLFH with prior bad outcomes choose to seek help elsewhere instead of hospitals or not at all, due to challenges in accessing health care in the rural areas as found in this study.

The level of PCC revealed in this study is higher in comparison to that found in three separate Nigerian studies, two in urban settings [8,10] and one in a semi-urban setting [11]. A possible explanation for this could lie in the differences in the socio-demographics of the study participants as explained above. The participants’ characteristics in the Southeastern Nigeria (urban) study were comparable to the participants in AKUH, N, a similar study setting, with a mean age of 30 years in both and study population of pregnant women seeking ANC. However, there were differences in the proportion of married women, those with at least a tertiary education, and primigravid in Southeastern Nigeria and AKUH, N [8]. Being married has been associated with increased utilization of maternal health services in different studies as previously discussed [15,17,20]. All women in the Southeastern Nigeria study were married but despite that, there was a lower level of PCC in comparison to AKUH, N women. This may point to an interplay of other factors such as socio-demographic characteristics of the partners and parity (more were multiparous in the Nigerian study) which was associated with less use of PCC from feeling well versed. More women in AKUH, N than Southeastern Nigeria sought PCC as routine, and this could be because more women in AKUH, N had at least a tertiary education which affects awareness levels. In both centers, however, this was the commonest reason for receiving PCC [8], which offers hope for increased PCC utilization with public education.

There were differences in the Northern Nigeria study (semi-urban) [11] compared to MLFH. The former studied women who had delivered within 24 months of the survey while in MLFH pregnant women attending ANC were studied. Therefore, there is a possibility of recall bias in the Northern Nigeria study since women had to recollect events that had happened over 24 months prior to data collection. There were differences in the mean age, the unemployed and those with at least a tertiary education in Northern Nigeria study and MLFH. All these factors showed significant associations with not only PCC utilization but also with other forms of maternal health services as discussed above and provide a possible explanation for the difference observed in the two studies. Another explanation for this could lie in the regional and country-wise differences in public health-seeking behaviors with Northern Nigeria having low demand for and utilization of maternal health services [21].

### 4.1. Strengths

This was a pilot study that compared the uptake and factors affecting PCC in both urban and rural settings, as well as in a private and a public hospital in Kenya. It is one of the very few studies conducted in Kenya in the area of PCC, which is an area that offers clear health benefits and is highly underutilized. This study may contribute to the bulk of PCC knowledge and awareness in order to inform policies and practices.

### 4.2. Weaknesses

The limitations of this study include the sample size that was not powered to allow firm conclusions to be drawn and the inability to analyze the role of some potentially confounding factors. There could also be a recall and reporting bias on the question of whether or not the participants received PCC from a health provider, given the variations in gestational age at enrollment. Further, these findings may not apply to urban women with different socio-demographic characteristics like those from the slums hence, a separate study in low resource urban areas may be helpful.

## 5. Conclusions

This is a first-of-its-kind pilot study into differences in PCC utilization in an urban and a rural health facility in Kenya. The main finding of this study was that the level of PCC in this setting, albeit low, is comparable to the global levels. This finding is interesting as both the WHO and the Ministry of Health in Kenya recognize the importance of PCC in policies aimed at reducing child mortality and improving maternal health [1,4,22]. In addition, there was a significant difference in the level of PCC utilization between the two study sites—lower in the rural rather than urban settings. This difference may be attributable to variance in the socio-demographic characteristics between the two study populations. Other factors may be at play here, such as differences in rural and urban demographics, infrastructure and access to resources, the differential impact of health policies and systems in urban and rural settings—this study did not look at these.

The small sample size of this study limits the strength of the conclusions about the determinants of PCC use in both settings. However, the findings are interesting and indicative. Further research is this field is therefore necessary in order to help inform strategies towards increasing PCC awareness and utilization in Kenya.

As a beginning of preconception care research in Kenya, this study can be considered as a pilot study and further research on the field is highly recommended.

## Figures and Tables

**Figure 1 ijerph-17-07430-f001:**
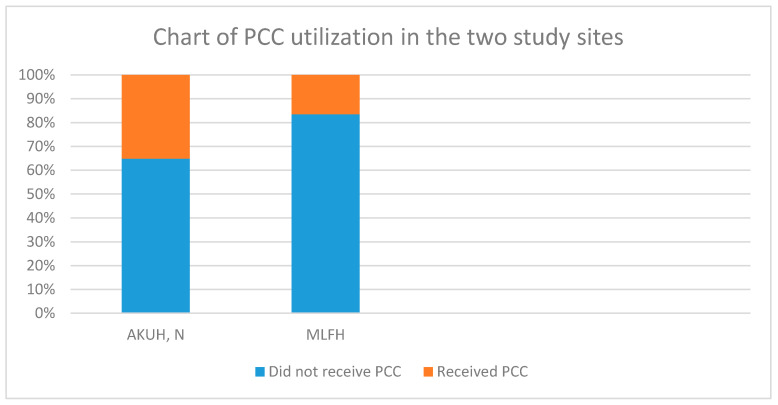
Preconception care (PCC) utilization in the two study sites.

**Table 1 ijerph-17-07430-t001:** Characteristics of study participants by site.

	AKUH, N *n* (%) or Mean ± S.D	Maragua *n* (%) or Mean ± S.D	*p*-Value (Chi-Square/*t*-Test)
**Age:**			
Mean	30.1 ± 4.0	26.8 ± 5.4	
<30 years	42 (43.3)	68 (70.1)	
≥30 years	55 (56.7)	29 (29.9)	<0.01
**Marital status:**			
Married	84 (86.6)	60 (61.9)	0.01
Single	12 (12.4)	31 (32.0)	
Divorced	0 (0.0)	1 (1.0)	
Widowed	1 (1.0)	5 (5.2)	
**Education:**			
Primary	0 (0.0)	18 (18.6)	<0.01
Secondary	3 (3.1)	54 (55.7)	
Tertiary	94 (96.9)	24 (24.7)	
None	0 (0.0)	1 (1.0)	
**Occupation:**			
Student	5 (5.2)	3 (3.1)	<0.01
Unemployed	8 (8.2)	21 (21.6)	
Farmer	0 (0.0)	21 (21.6)	
Business	17(17.5)	33 (34.0)	
Professional	67 (69.1)	19 (19.6)	
**Parity:**			
Primigravida	33 (34.0)	38 (39.2)	0.46
Multigravida	64 (66.0)	59 (60.8)	
**Prior poor pregnancy outcomes:** Yes No Miscarriages Molar pregnancy Ectopic pregnancy Preterm birth Stillbirth Birth defects	(*n* = 64) 43 (67.2) 21 (32.8) 21 (32.8) 1 (1.6) 1 (1.6) 5 (7.8) 2 (3.1) 0 (0.0)	(*n* = 59) 34 (57.6) 25 (42.4) 17 (28.8) 0 (0.0) 6 (10.2) 3 (5.1) 4 (6.8) 2 (3.4)	0.19
Caesarian section delivery	(*n* = 64)	(*n* = 59)	0.28
	16 (25.0)	17 (28.8)	
Medical conditions during prior pregnancy (such as diabetes, hypertension, cholestasis in pregnancy etc.)	(*n* = 64)	(*n* = 59)	0.95
	4 (6.3)	8 (13.6)	
**Preexisting medical conditions**			
Diabetes	1 (1.0)	3 (3.1)	0.23
High blood pressure	2 (2.1)	1 (1.0)	
Heart disease	0 (0.0)	0 (0.0)	
Asthma	4 (4.1)	5 (5.2)	
Epilepsy	1 (1.0)	1 (1.0)	
HIV	4 (4.1)	8 (8.2)	
None	85 (87.6)	79 (81.4)	

**Table 2 ijerph-17-07430-t002:** Table of health-seeking behaviour by site.

	AKUH, N *n* (%) or Mean ± S.D	Maragua *n* (%) or Mean ± S.D	*p*-Value (Chi-Square/*t*-Test)
**Last time seen by health professional other than for ANC**			
Within 3 months before pregnancy	43 (44.3)	11 (11.3)	<0.01
Within 6 months before pregnancy	16 (16.5)	6 (6.2)	
Within 12 months before pregnancy	7 (7.2)	8 (8.2)	
More than a year before pregnancy	17 (17.5)	38 (39.2)	
Never	14 (14.4)	34 (35.1)	
**Purpose of visit to health care professional**			
Regular health check up	25 (25.8)	1 (1.0)	<0.01
Sick and needed medical care	35 (36.1)	54 (55.7)	
To discuss pregnancy plans and preparation	23 (23.7)	8 (8.2)	
Not applicable (i.e., had never seen health professional before)	14 (14.4)	34 (35.1)	
**Received PCC:** Yes	34 (35.1)	16 (16.5)	<0.01
No	63 (64.9)	81 (83.5)	
**Sought PCC elsewhere other than from health provider:**			
No	60 (61.9)	71 (73.2)	0.19
Yes	37 (38.1)	26 (26.8)	
Family	16 (16.5)	12 (12.4)	
Friend	16 (16.5)	18 (18.6)	
Radio/ Television	0 (0.0)	0 (0.0)	
Social media	8 (8.2)	4 (4.1)	
Traditional healer	0 (0.0)	0(0.0)	
Other (Internet, google)	8 (8.2)	3 (3.1)	
**Challenges to accessing medical care**			
Distance	2 (2.1)	2 (2.1)	<0.01
Financial	4 (4.1)	28 (28.9)	
Other (time)	2 (2.1)	0 (0.0)	
None	89 (91.8)	67 (69.1)	
**About becoming pregnant**			
Right time	79 (81.4)	59 (60.8)	<0.01
Okay, but not right time	17 (17.5)	29 (29.9)	
Wrong time	1 (1.0)	9 (9.3)	
**Possible to prepare for pregnancy**			
Yes	91 (93.8)	65 (67.0)	<0.01
No	6 (6.2)	32 (33.0)	
**There is enough information about PCC**			
Yes	38 (39.2)	6 (6.2)	<0.01
No	59 (60.8)	91 (93.8)	

**Table 3 ijerph-17-07430-t003:** PCC utilization.

Characteristic (*n* = Number of Respondents)	PCC Received (%)	Odds Ratio (95% Confidence Interval)	*p*-Value
**Age:** <30 years	20/110 (18.1)		
≥30 years	30/84 (35.7)	2.5 (1.3–4.8)	0.01
**Site:** AKUH, N	34/97 (35.1)		
MLFH	16/97 (16.5)	0.3 (0.2–0.7)	<0.01
**Marital status:** Not Married	7/50 (14.0)		
Married	43/144 (29.9)	2.6 (1.1–6.3)	0.03
**Education:** Below tertiary	9/76 (11.4)		
Tertiary	41/118 (34.7)	4.0 (1.8–8.8)	<0.01
**Occupation:**			
No formal employment	16/108 (14.8)		
Formal employment	34/86 (39.5)	3.8 (1.9–7.5)	<0.01
**Parity:** Primigravida	10/71 (14.1)		
Multigravida	40/123 (32.5)	3.0 (1.4–6.3)	0.01
**Prior poor pregnancy outcomes**			
No	10/17 (58.8)	3.1 (1.6–6.0)	<0.01
Yes	30/106 (28.3)		
Miscarriages	26/38 (42.1)	2.6 (1.2–5.5)	0.01
Preterm birth	3/8 (37.5)	1.8 (0.4–7.7)	0.44
Stillbirth	2/6 (33.3)	1.5 (0.3–8.2)	0.67
Birth defects	1/2 (50.0)	2.9 (0.2–47.5)	0.43
**Preexisting medical conditions:**			
No	42/164 (25.6)		
Yes	8/30 (26.7)	1.1 (0.4–2.3)	0.90
**Sought PCC elsewhere (not health provider):**			
No	23/131 (17.6)		
Yes	27/63 (42.9)	3.5 (1.8–6.9)	<0.01
**Challenges to accessing medical care:**			
No	40/156 (25.6)		
Yes	10/38 (26.3)	1.0 (0.5–2.3)	0.94
**About becoming pregnant**			
Not right time	8/56 (14.3)		
Right time	42/138 (30.4)	2.6 (1.1–6.0)	0.02
**There is enough information about PCC:**			
No	34/150 (22.7)		
Yes	16/44 (36.4)	2.0 (0.9–4.0)	0.07

**Table 4 ijerph-17-07430-t004:** Multivariate logistic regression.

Variables	Adjusted Odds Ratio (95% Confidence Interval)	*p*-Value
Site	1.32 (0.479–3.612)	0.60
Age	1.18 (0.552–2.511)	0.67
Marital status	1.58 (0.587–4.227)	0.37
Education	3.08 (0.955–9.940)	0.06
Occupation	2.20 (0.951–5.083)	0.07
Parity	1.68 (0.607–4.652)	0.32
Prior poor pregnancy outcomes	2.39 (0.978–5.861)	0.06

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
