# Peer review of "Preconception Care among Pregnant Women in an Urban and a Rural Health Facility in Kenya: A Quantitative Study"

_ijerph, 2020, doi:10.3390/ijerph17207430_

Round 1

Reviewer 1 Report

This study was undertaken to determine the level of utilization and causes for putative differences in preconception care (PCC) use between an urban and a rural setting in Kenya. A questionnaire was administered to consecutive pregnant women attending their antenatal visit in Mother Child Health Clinics at either a single private tertiary health facility or a public rural district hospital. Descriptive statistics were drawn from participant answers, and univariate and multivariate analyzes of data were performed to identify factors that may influence the use of PCC between and within sampled groups. Differences were found for PCC level of utilization between rural and urban participants.

Studies that increase knowledge of circumstances that lead to designing strategies aimed towards a positive impact in fetal and maternal welfare are vital in every community. The main limitation of this work is however, as authors acknowledge, the lack of power to appropriately analyze associations between PCC use and independent variables, as well as the effects and adjustment of potential confounding factors. The limited number of communities, facilities, and participants sampled reduced the heterogeneity in the sampled population and thus hindered the power of the analyzes and the reach of the findings. Hence, significant determinants of PNC use could not be established.

Also, the hospital sampled in the urban setting is a referral tertiary hospital which may have influenced the  characteristics of attending patients, as shown for example by differences in participants age groups, where older women that may have an increased pregnancy risk could be more likely to attend a tertiary level hospital for appropriate care. This could have created biased results. The cluster design effect should have been considered in the experimental design to reduce bias in obtained results.

In addition, information in range of population within geographical contexts is lacking.

Moreover, the study does not represent all regions or conditions of Kenya. From the title the reader can be misled in thinking that the study encompasses several rural and urban settings, overstating its reach and possible inference.

Other specific comments:

-Title: It should be reformulated to better represent the reach of the study and its findings.

- L18 and L92-93 For readers outside Kenya, it would be desirable to indicate general level of care of  “a rural level four hospital”, since it does not seem to fit within a more common health system classification (which includes primary, secondary and tertiary levels of care).

-L 103 separate “at least”

- L104 which were the criteria used to establish that it was in effect a “good representation” of the population? Was the only considered factor that “97% of pregnant women have at least one ANC consult”? If so, this standard should be revised and improved to avoid result bias.

-L106-111 were non-response ratio, recollection bias and cluster effect considered in calculation of sample size?

L112-122 Validation of the questionnaire is important. Presentation of the complete questionnaire as supplementary information could be useful for interested readers.

Fig 1. Delete. Not necessary. Information is sufficiently explained within the text and easy to follow.

Author Response

Open Review

English language and style

( ) Extensive editing of English language and style required
( ) Moderate English changes required
(x) English language and style are fine/minor spell check required
( ) I don't feel qualified to judge about the English language and style

Yes

Can be improved

Must be improved

Not applicable

Does the introduction provide sufficient background and include all relevant references?

(x)

( )

( )

( )

Is the research design appropriate?

( )

( )

(x)

( )

Are the methods adequately described?

( )

(x)

( )

( )

Are the results clearly presented?

(x)

( )

( )

( )

Are the conclusions supported by the results?

( )

( )

(x)

( )

Comments and Suggestions for Authors

This study was undertaken to determine the level of utilization and causes for putative differences in preconception care (PCC) use between an urban and a rural setting in Kenya. A questionnaire was administered to consecutive pregnant women attending their antenatal visit in Mother Child Health Clinics at either a single private tertiary health facility or a public rural district hospital. Descriptive statistics were drawn from participant answers, and univariate and multivariate analyzes of data were performed to identify factors that may influence the use of PCC between and within sampled groups. Differences were found for PCC level of utilization between rural and urban participants.

Studies that increase knowledge of circumstances that lead to designing strategies aimed towards a positive impact in fetal and maternal welfare are vital in every community. The main limitation of this work is however, as authors acknowledge, the lack of power to appropriately analyze associations between PCC use and independent variables, as well as the effects and adjustment of potential confounding factors. The limited number of communities, facilities, and participants sampled reduced the heterogeneity in the sampled population and thus hindered the power of the analyzes and the reach of the findings. Hence, significant determinants of PNC use could not be established.

Also, the hospital sampled in the urban setting is a referral tertiary hospital which may have influenced the  characteristics of attending patients, as shown for example by differences in participants age groups, where older women that may have an increased pregnancy risk could be more likely to attend a tertiary level hospital for appropriate care. This could have created biased results. The cluster design effect should have been considered in the experimental design to reduce bias in obtained results.

In addition, information in range of population within geographical contexts is lacking. Moreover, the study does not represent all regions or conditions of Kenya. From the title the reader can be misled in thinking that the study encompasses several rural and urban settings, overstating its reach and possible inference.

It is true that the urban study setting had more women above 30 years compared to the rural setting. However, this is not because it is a tertiary facility but rather a private facility whose accessibility is limited to those who can afford the care. This therefore means that a majority of women who seek care from this facility are older, working class women. The mean age was 30+/-4 versus 26.8+/-5.4 and this difference was significant but does not have a great impact on reproductive outcomes as most reproductive concerns are associated with women in their late thirties and forties. It is therefore likely that most women who sought care at AKUH, N did it just because they could afford and access, not because of higher pregnancy risk

Other specific comments:

-Title: It should be reformulated to better represent the reach of the study and its findings.

  • We have reformulated the title…..PRECONCEPTION CARE AMONG PREGNANT WOMEN IN AN URBAN AND A RURAL HEALTH FACILITY IN KENYA: A QUANTITATIVE STUDY. (L2,3)

- L18 and L92-93 For readers outside Kenya, it would be desirable to indicate general level of care of  “a rural level four hospital”, since it does not seem to fit within a more common health system classification (which includes primary, secondary and tertiary levels of care).

  • Level four hospitals are the first referral hospitals in Kenya, with both outpatient and inpatient health services and referral services. They have the following clinical services which are run by either an on-site specialist in the field or a specialist who covers several of the level four hospitals within the region: obstetrics and gynaecology; child health; medicine; surgery and anaesthesia (this has been added in text). (L96-100)

-L 103 separate “at least”

  • Done (L114).

- L104 which were the criteria used to establish that it was in effect a “good representation” of the population? Was the only considered factor that “97% of pregnant women have at least one ANC consult”? If so, this standard should be revised and improved to avoid result bias.

  • (L113-117) In both counties, more than 97% of pregnant women seek ANC in a health facility. Sampling from this population gave a good representation of pregnant women in the two counties and also a presentation on people who needed PCC (whether they used it or not) as they are already pregnant. Sampling reproductive aged women overall, in as much as PCC applies to them as well, may not represent all those who actually need PCC as some of them are not interested in pregnancies/children at all and some have reached their desired family size and no longer need PCC.

-L106-111 were non-response ratio, recollection bias and cluster effect considered in calculation of sample size?

  • These elements were not considered in the sample size calculation.

L112-122 Validation of the questionnaire is important. Presentation of the complete questionnaire as supplementary information could be useful for interested readers.

  • Questionnaires were validated by conducting a pilot study as explained in text (L135-140). The questionnaire will be sent as supplementary information as requested.

Fig 1. Delete. Not necessary. Information is sufficiently explained within the text and easy to follow.

  • Figure 1 Deleted.

 Submission Date

21 July 2020

Date of this review

31 Jul 2020 02:21:39

Reviewer 2 Report

Preconception Care is very important for pregnant women. Improvement of PC is necessary around the world. Joan Okemo1 et al investigated the difference in the Level of Utilization and Determinants of Preconception Care between Pregnant Women in Urban and Rural Settings in Kenya, which provided local reference. However, due to small sample size and nature of exploration, it was little difficult to make the conclusion generalized to general population. The following issues should be addressed further.

  1. Due to two selected hospitals as studying site, selection bias might occur. So conclusion should be cautious.
  2. Present estimated sample size could have a lower power to test the difference between urban and rural pregnant women in related index besides PCC.
  3. It was not clear about definition of PCC, authors should provide clear definition.
  4. Authors did not provide information on the gestational age for those two groups. Maybe gestational age could affect recall of PCC.
  5. Authors should provide detailed covariates which could be considered in Multivariate logistic regression and the reasons why to be selected. Due to small sample size, the covariates considered in model are not suggested to be grouped more than two categories, which could be one reason that not any significant results were found in table 4.

Author Response

Open Review

English language and style

( ) Extensive editing of English language and style required
( ) Moderate English changes required
( ) English language and style are fine/minor spell check required
(x) I don't feel qualified to judge about the English language and style

Yes

Can be improved

Must be improved

Not applicable

Does the introduction provide sufficient background and include all relevant references?

( )

(x)

( )

( )

Is the research design appropriate?

( )

( )

(x)

( )

Are the methods adequately described?

( )

( )

(x)

( )

Are the results clearly presented?

( )

( )

(x)

( )

Are the conclusions supported by the results?

( )

( )

(x)

( )

Comments and Suggestions for Authors

Preconception Care is very important for pregnant women. Improvement of PC is necessary around the world. Joan Okemo1 et al investigated the difference in the Level of Utilization and Determinants of Preconception Care between Pregnant Women in Urban and Rural Settings in Kenya, which provided local reference. However, due to small sample size and nature of exploration, it was little difficult to make the conclusion generalized to general population. The following issues should be addressed further.

  1. Due to two selected hospitals as studying site, selection bias might occur. So conclusion should be cautious.
  • Conclusion has been modified. See track changes in article (L306-315).
  • Conclusion: The main finding of this study was that the level of PCC in both an urban and a rural health facility in Kenya, albeit low, is comparable to the global levels. This, just like in many parts of the world, calls for strategies to be put in place towards increasing PCC awareness and utilization.
  • The difference in level of PCC between rural and urban participants observed in this study could have arisen from the observed differences in the socio-demographic characteristics in the urban and rural populations. Other non-patient related factors such as differences in rural and urban infrastructure, resources, local communities, existing health policies and systems could also account for the difference observed in this study; however, the study did not look at these.
  1. Present estimated sample size could have a lower power to test the difference between urban and rural pregnant women in related index besides PCC.
  • This limitation was well noted in the article (L302-303).
  1. It was not clear about definition of PCC, authors should provide clear definition.
  • (L77-81)The study definition of PCC utilization was contact with any health provider before current pregnancy and having discussed about pregnancy planning and preparation (see opening statement in Materials and methods section).
  • To elicit the level of PCC utilization, we used question 3 in section four of the questionnaire (attached) which stated, “I talked about pregnancy plans and preparation with a health care provider before I got pregnant.”
  1. Authors did not provide information on the gestational age for those two groups. Maybe gestational age could affect recall of PCC.
  • Well noted. Information on gestational age was not captured and it is true a recall bias could arise from this which is among our study limitations (L303-304).
  1. Authors should provide detailed covariates which could be considered in Multivariate logistic regression and the reasons why to be selected. Due to small sample size, the covariates considered in model are not suggested to be grouped more than two categories, which could be one reason that not any significant results were found in table 4.
  • (L204-211) Please see this introduction paragraph to table 4 which provides information on which variables were subjected to multivariate analysis and why….“Univariate analysis revealed a strong association between PCC and age, site, marital status, education level, occupation, parity, prior poor pregnancy outcomes and seeking PCC elsewhere, and these were subjected to multivariate logistic regression to define the adjusted odds ratios. Because of low sample size the confidence intervals were wide and no significant difference below the 0.05 level were found, but trends were observed for education and prior poor pregnancy outcomes”…..

Submission Date

21 July 2020

Date of this review

23 Jul 2020 11:26:46

Reviewer 3 Report

Dear Authors. Congratulations on the study.
Abstract: Should be structured. Please pay attention to the Key Words, some are not MESH TERMS or descriptors in health science. You can check for the English Language here: http://decs.bvs.br/I/homepagei.htm
The introduction is correct.
However Material and Methods should be improved. First of all, the sample size is not so clear and it can generate misunderstandings. I suggest clarifying the process of the sample and the inclusion criteria. Besides, 95 IC with 80 % is in the limit of precision.
The inclusion criteria are not so clear because you don´t specify clearly.
On the other hand, the tools are developed and based on the study objectives and literature review, but we have not any properties of these tools?? Do we know the psychometric adjustment of the questionnaire?
Concerning the statistics analysis please focus that multivariate logistic regression in some items is not significant. Please focus on the strength of the study, not in the weakness
Discussion: You should improve. Results should be explained in the parts of the result. Not in the discussion. However, you can use the discussion to try to relate your results with the evidence.

Author Response

Open Review

English language and style

( ) Extensive editing of English language and style required
( ) Moderate English changes required
(x) English language and style are fine/minor spell check required
( ) I don't feel qualified to judge about the English language and style

Yes

Can be improved

Must be improved

Not applicable

Does the introduction provide sufficient background and include all relevant references?

(x)

( )

( )

( )

Is the research design appropriate?

( )

( )

(x)

( )

Are the methods adequately described?

( )

(x)

( )

( )

Are the results clearly presented?

(x)

( )

( )

( )

Are the conclusions supported by the results?

( )

(x)

( )

( )

Comments and Suggestions for Authors

Dear Authors. Congratulations on the study.

Abstract: Should be structured.

  • (L11-25) I followed the writing guidelines of IJERPH which specified doing a paragraph of abstract without breaking it into sections.

Please pay attention to the Key Words, some are not MESH TERMS or descriptors in health science. You can check for the English Language here: http://decs.bvs.br/I/homepagei.htm

  • (L26) Keywords edited as shown: Preconception care; Kenya; Urban; Rural; Level; Determinants; Utilization.

The introduction is correct.
However Material and Methods should be improved. First of all, the sample size is not so clear and it can generate misunderstandings. I suggest clarifying the process of the sample and the inclusion criteria. Besides, 95 IC with 80 % is in the limit of precision.

  • (L118-133) Please see the track changes made in the article in this section. More information on sample size calculation has been provided, including the formula, the 95 CI and 80% power which are what we used.

The inclusion criteria are not so clear because you don´t specify clearly.

  • (L110-112) This was the inclusion criteria provided in the article…”The participants were eligible if they were aged ≥18 years, able to speak English or Kiswahili languages and were attending their antenatal visit during current pregnancy, regardless of the gestational age.”

On the other hand, the tools are developed and based on the study objectives and literature review, but we have not any properties of these tools?? Do we know the psychometric adjustment of the questionnaire?

  • (L141-146) This section summarizes the questionnaire components. I have attached a copy of the questionnaire as supplemental information….” A 5-10 minutes self-administered questionnaire was used to collect data including socio-demographics section (age, marital status, education, occupation, residential area); obstetric history (parity, gestational age in weeks by last menstrual period (LMP) or first trimester scan and prior obstetrics outcomes); pre-existing medical conditions and a section with questions to determine the level of PCC prior to current pregnancy and some more determinants”
  • On psychometric adjustments, we did conduct a pilot study to validate the data collection tools as captured in this paragraph (L135-140)….” They were thereafter validated on 8 antenatal women in AKUH, N and MLFH MCH clinics to test the idioms of the languages used, response time, check whether the questions were appropriately framed, inoffensive, clear, easy to understand, able to elicit discussions and address the intended questions for the study. They were found to be suitable for the study and did not require any adjustments….

Concerning the statistics analysis please focus that multivariate logistic regression in some items is not significant.

  • (L209-211) Please see the track changes on the article as follows….”Because of low sample size the confidence intervals were wide and no significant difference below the 0.05 level were found, but trends were observed for education and prior poor pregnancy outcomes.”

Please focus on the strength of the study, not in the weakness

  • Please see the following changes in the strengths and weaknesses section…. (L297-304)

Strengths

This study compared the uptake and factors affecting PCC in both urban and rural settings, as well as in a private and a public hospital in Kenya. It is one of the very few studies in the area of PCC that has clear health benefits and is highly underutilized, and may contribute to the bulk of knowledge and awareness to inform policies and practices.

Weaknesses

Limitations include the sample size that was not powered to allow firm conclusions and analyze the role of some potentially confounding factors, as well as recall and reporting bias given the variations in gestational age at enrollment.

Discussion: You should improve. Results should be explained in the parts of the result. Not in the discussion. However, you can use the discussion to try to relate your results with the evidence.

  • Well noted. See track changes in the article under discussion section. The results have been removed from discussion section. (L215-295)

Submission Date

21 July 2020

Date of this review

14 Aug 2020 12:19:02

Round 2

Reviewer 1 Report

I appreciate the authors' effort to improve the manuscript, as well as to consider the suggestions that were previously sent. However, the main shortcoming of the study, which lies in its methodology, was not sufficiently addressed. Since this may lead to biased results and conclusions, it is my opinion that this paper is still lacking the needed scientific soundness to appropiately convey significance of observed circumstances and limitations.

Author Response

  • Well noted. Re-framed the study as a pilot study as per the academic editors’ recommendations (L13 and L318-319).

Reviewer 2 Report

authors have addressed most of my comments and the manuscript has been improved.

Author Response

  • Thank you. Well noted.

Reviewer 3 Report

Dear Authors. With the modifications suggested the article from my point of view can be published. Please provide a high resolution in the formula for the size calculation.

Author Response

  • Thank you. High resolution for formula size provided as below…(L124-126)

  • N=2× (Zα+Zβ)2 ×    P1 (1-P1) + P2(1-P2)

                                                 (P1-P2)2